# Molecular Regulatory Mechanism of Exogenous Hydrogen Sulfide in Alleviating Low-Temperature Stress in Pepper Seedlings

**DOI:** 10.3390/ijms242216337

**Published:** 2023-11-15

**Authors:** Xueping Song, Li Zhu, Dong Wang, Le Liang, Jiachang Xiao, Wen Tang, Minghui Xie, Zhao Zhao, Yunsong Lai, Bo Sun, Yi Tang, Huanxiu Li

**Affiliations:** College of Horticulture, Sichuan Agricultural University, Chengdu 611130, China; 17793496171@163.com (X.S.); 18784009622@163.com (L.Z.); wd814702541@163.com (D.W.);

**Keywords:** pepper (*Capsicum annuum* L.), low-temperature stress, hydrogen sulfide, transcriptional regulation

## Abstract

Pepper (*Capsicum annuum* L.) is sensitive to low temperatures, with low-temperature stress affecting its plant growth, yield, and quality. In this study, we analyzed the effects of exogenous hydrogen sulfide (H_2_S) on pepper seedlings subjected to low-temperature stress. Exogenous H_2_S increased the content of endogenous H_2_S and its synthetase activity, enhanced the antioxidant capacity of membrane lipids, and protected the integrity of the membrane system. Exogenous H_2_S also promoted the Calvin cycle to protect the integrity of photosynthetic organs; enhanced the photosynthetic rate (Pn), stomatal conductance (Gs), transpiration rate (Tr), and photosynthesis; and reduced the intercellular CO_2_ concentration (Ci). Moreover, the activities of superoxide dismutase, peroxidase, catalase, and anti-cyclic glutathione (ASA-GSH) oxidase were improved to decompose excess reactive oxygen species (ROS), enhance the oxidative stress and detoxification ability of pepper seedlings, and improve the resistance to low-temperature chilling injury in ‘Long Yun2’ pepper seedlings. In addition, the H_2_S scavenger hypotaurine (HT) aggravated the ROS imbalance by reducing the endogenous H_2_S content, partially eliminating the beneficial effects of H_2_S on the oxidative stress and antioxidant defense system, indicating that H_2_S can effectively alleviate the damage of low temperature on pepper seedlings. The results of transcriptome analysis showed that H_2_S could induce the MAPK-signaling pathway and plant hormone signal transduction; upregulate the expression of transcription factors WRKY22 and PTI6; induce defense genes; and activate the ethylene and gibberellin synthesis receptors ERF1, GDI2, and DELLA, enhancing the resistance to low-temperature chilling injury of pepper seedlings. The plant–pathogen interaction was also significantly enriched, suggesting that exogenous H_2_S also promotes the expression of genes related to plant–pathogen interaction. The results of this study provide novel insights into the molecular mechanisms and genetic modifications of H_2_S that mitigate the hypothermic response.

## 1. Introduction

Low temperatures remain a major challenge for plant growth and survival since they impact the productivity of important cash crops [1]. Low temperatures have an impact on different growth stages of rice, such as the booting stage, flowering stage, and filling stage, thus reducing the rice yield and affecting food security [2]. Low temperatures can also affect the degree of stomatal opening and closure on the leaf surface, thereby affecting CO_2_ absorption, destroying the chloroplast thylakoid membrane, affecting the photosystem reaction center, and inhibiting the activity of key enzymes in the Calvin cycle [3,4]. Researchers found that low temperatures can cause an imbalance between reactive oxygen species (ROS) production and scavenging. If not detoxified in time, excess ROS increase the membrane permeability and membrane lipid peroxidation, DNA damage, and protein denaturation [5]. To maintain a ROS balance, the defense system of antioxidant enzymes superoxide dismutase (SOD); peroxidase (POD); catalase (CAT); and the ascorbate glutathione cycle (ASA–GSH) and its key enzymes, such as ascorbate peroxidase (APX), monodehydroascorbate reductase (MDHAR), and dehydroascorbate reductase (DHAR), are activated [6,7]. Increasing the content of endogenous reducing glutathione (GSH) and the ratio of GSH to oxidizing glutathione (GSH/GSSG) can alleviate the decrease in photosynthetic parameters and the destruction of photosynthetic apparatus caused by low temperatures. This will also increase the key photosynthetic enzymes Ribulose diphosphate carboxylase/oxidase (RubisCO), sepullose-1, 7-diphosphatase (SBPase), and Fructose-1, 6-diphosphate aldolase (FBA) activities [8].

The molecular regulatory network was previously studied by mining transcription factors and genes that respond to low-temperature stress. The cold regulation pathway ICE (inducer of CBF expression) CBF (C repeat binding factor)-COR (cold-regulated) is the most understood defense mechanism in response to cold stress [9]. Transcriptional and metabolic analyses of pepper and bell pepper showed that most differentially expressed genes (DEGs) are involved in amino acid biosynthesis, plant hormone signal transduction, RNA groups, and mitogen-activated protein kinase (MAPK) signaling pathways [10,11,12]. Six transcription factor (TF) families have been identified, namely, the *AP2*/*ERF*, *C2H2*, *WRKY*, *bHLH*, *NAC*, and *MYB*-related families, and differential genes have been classified into the *NAC* family and *WRKY* family using quantitative real-time PCR (qRT–PCR) verification [13,14]. *WRKY* proteins, which comprise a superfamily of transcription factors, are involved in plant low-temperature responses [12]. During crop cultivation, H_2_S fumigation, irrigation, mulching film, and greenhouse cultivation are often used to protect against the damage caused by low temperatures.

Hydrogen sulfide (H_2_S) is a small signaling molecule involved in seed germination, plant growth and development, and the regulation of stress tolerance [15]. Exogenous H_2_S can regulate the ability of plants to maintain the K^+^/Na^+^ balance under salt stress [16] and improve salt and drought tolerance by increasing antioxidant enzyme activity, ASA–GSH cycle homeostasis, and endogenous H_2_S content [17,18]. H_2_S can work synergistically with other plant hormones/signaling molecules, such as CO, Ca^2+^, SA, and ABA, to improve the heat resistance of plants, alleviate oxidative stress caused by abiotic stress, promote proline accumulation and antioxidant activity, and reduce electrolyte leakage and H_2_O_2_ and MDA contents [19]. Studies on the role of H_2_S in enhancing cold tolerance were conducted for various plants. For example, in cold tolerance studies on cucumbers and blueberries, H_2_S increased the activity of L-D-cysteine desulfhydrase (CDes) and promoted the catalytic synthesis of endogenous H_2_S [20]. 

Pepper (*Capsicum annuum* L.) is an important thermophilic vegetable with an optimal growth temperature range of 21–28 °C [21]. *Capsicum* is very sensitive to low temperatures, and low-temperature stress can seriously affect its growth and development, limiting its yield and quality [12]. Many exogenous substances, such as brassinolide (Br), salicylic acid (SA), and 5-aminolevulinic acid (ALA), can alleviate the stress caused by low temperatures in capsicum [22,23,24]. The effect of H_2_S on low-temperature stress in pepper mostly remains at the physiological level, and data on the signaling pathway in response to low-temperature stress remain limited. Therefore, in this study, the relationships between the physiological characteristics, molecular response mechanisms, and signaling pathways of exogenous H_2_S on pepper seedlings under low-temperature stress were investigated to better facilitate the cultivation and screening of high-quality cultivars.

## 2. Results

### 2.1. Phenotype and Membrane System Changes after Low-Temperature Stress

The phenotype after low-temperature stress (Figure 1A) showed that with the extension of low-temperature stress, compared with CK, other treatment leaves showed obvious water loss and wilting; the degree of wilting and damage was manifested as LHT > LCK > LHS > CK. Compared with CK and LCK, the contents of REL and MDA in the LHS treatment were significantly reduced. At 24 h, the content of MDA in the LHS treatment was significantly reduced by 60.34% compared with the LCK treatment, and the LHT treatment increased the REL (Figure 1B,C), and for 12–48 h, LHT treatment increased the MDA content. These results indicate that H_2_S enhanced the stability of the membrane system by reducing the content of REL and MDA, maintaining the integrity of the plant phenotype.

### 2.2. Effects of Low-Temperature Stress on the Photosynthetic Index and Calvin Cycle Enzymes

Compared with CK, with increasing stress time, Pn, Gs, and Tr decreased continuously, whereas Ci increased during 0–24 h and decreased during 24–48 h. Exogenous H_2_S increased the Pn, Gs, and Tr while decreasing Ci (Figure 2A–D). Compared with CK, LHS significantly increased Pn (26.9%), Gs (35.6%), and Tr (65.5%), but decreased Ci (17.8%) at 0 h. At 24 h, compared with LCK, LHS significantly increased Pn, Gs, and Tr (66.49%, 44.2%, and 39.96%, respectively), while Ci significantly decreased by 29.4% (Figure 2A,B,D). This indicates that H_2_S can increase the Pn and Gs of pepper seedlings under low-temperature stress and decrease Ci and Tr, maintaining the stability of the photosynthetic system and improving the resistance to low-temperature chilling injury in pepper seedlings.

Compared with CK, the activity of RubisCO decreased under low-temperature stress, while that of FBAase and SPBase first decreased and then increased. Compared with CK and LCK at 0 h, LHS treatment significantly increased RubisCO and FBAase by 10.3% and 16.6% and by 25.9% and 15.6%, respectively. With increasing stress time, compared with LCK, the LHS treatment significantly increased RubisCO and FBAase, which peaked at 12 h (25.8% and 23.0%, respectively). However, LHT significantly reduced the enzyme activity (Figure 2E–G). These results indicate that H_2_S can reduce the inhibition of the photosynthetic index under low-temperature stress, increase the activity of key photosynthetic enzymes, and maintain the stability of light capture and other systems under low-temperature stress.

### 2.3. Effects of Low-Temperature Stress on ROS and Enzymatic Oxidation

With the extension of the low-temperature stress time, compared with the control, the Pro content of the LHS treatment increased significantly, the O_2_^−^ content decreased steadily but did not reach a significant level, and the H_2_O_2_ content decreased significantly. The Pro content of the LCK treatment also showed the same trend, but the O_2_^−^ and H_2_O_2_ content increased significantly only at 24 h and 48 h, and the H_2_O_2_ content increased significantly only at 0 h and 48 h, while the Pro, O_2_^−^, and H_2_O_2_ contents increased significantly due to the LHT treatment. Compared with CK and LCK, the contents of O_2_^−^ and H_2_O_2_ in the LHS treatment were significantly decreased at 24 h (CK: 10.17% and 36.45%, respectively; LCK: 19.23% and 21.83%, respectively; Figure 3B,C). LHS increased the Pro content by 70% compared with CK at 0 h. At low temperatures, LHS significantly increased the Pro content (Figure 3A). 

Under low-temperature stress, the SOD activity of the LCK treatment was significantly increased at 24 h and significantly decreased at other times compared with the control. The POD activity increased significantly at 0 h; decreased significantly at 6 h, 12 h, and 48 h; and was not significant at 24 h. The CAT activity increased significantly at all times except 0 h, and showed a trend of increasing first and then decreasing. Compared with the control group, the SOD and POD activities in the LHS treatment first increased and then decreased (Figure 3D,E), and the CAT activities first decreased and then increased (Figure 3F). In the LHT treatment, the SOD activity at 48 h and CAT activity at 24 h were significantly decreased and were significantly increased at other times. Compared with LCK, the LHS treatment significantly increased the enzyme activity and reached the peak at 24 h, SOD increased by 27.28%, POD increased by 267%, and CAT increased by 125%. In this study, exogenous H_2_S improved the enzyme activity by initiating the enzymatic oxidation system, and the enzyme activity was strongest at low temperatures for 24 h, increasing the content of Pro, removing excess ROS, and enhancing the low-temperature chilling injury of pepper seedlings.

### 2.4. Effects of Low-Temperature Stress on the ASA–GSH Cycle

H_2_S promoted the enzymatic activity of the ASA–GSH circulatory system, improved the metabolic function of plants, and reduced the accumulation of reactive oxygen species in cells, thereby reducing the damage caused by low-temperature stress in the pepper seedlings. Compared with the control, the activity of MDHAR, which is a key enzyme in the ASA-GSH cycle, decreased first and then increased after low-temperature stress, and the enzyme activities of APX and MDHAR first increased and then decreased, and further decreased again after reaching a peak value (Figure 4A–C). Compared with CK and LCK, the LHS treatment significantly increased the enzyme activity, and APX peaked at 24 h, increasing by factors of 500% and 240%, respectively. LHS only significantly increased the ASA/DHA ratio at 6 h. Compared with LCK, the LHS treatment significantly reduced the DHA content, and increased the ASA/DHA ratio by 160% at 6 h (Figure 4D–F). The LHS treatment significantly increased the contents of GSH and GSSG by 94.88% and 36.01% at 24 h, respectively, and significantly increased the ratio of 24 h GSH/GSSG by 91.94%, which was the peak value (Figure 4G–I). Exogenous H_2_S significantly increased the contents of GSH and GSSG, and the ratio of GSH/GSSG in the ASA–GSH cycle after the low-temperature stress significantly increased the activities of the key enzymes MDHAR, DHAR, and APX, and peaked at 24 h, which enhanced the cold tolerance of the pepper seedlings.

### 2.5. Effects of Endogenous H_2_S and Related Synthases

Exogenous H_2_S increased the content of endogenous H_2_S and the activity of its key synthase in pepper seedlings under low-temperature stress. Conversely, the endogenous H_2_S content increased with increasing stress time (Figure 5A), peaking at 12 h and then gradually decreasing (but still significantly higher than that in CK). Compared with CK and LCK, the LHS treatment significantly increased the content of endogenous H_2_S and the activities of the LCD and DCD (CK: 243%, 143%, and 228%, respectively; LCK: 13.38%, 39.45%, and 47.07%, respectively) in the leaves at 24 h. The LHT treatment significantly reduced the endogenous H_2_S and enzyme activity (Figure 5B,C).

### 2.6. RNA Sequencing, Assembly, and Analysis of Differential Genes

The 12 cDNA libraries were sequenced: the average amount of clean data was 76.83 GB and the amount of clean data of each sample was 5.70 GB. The 12 cDNA libraries were sequenced: the average amount of clean data was 76.83 GB, the amount of clean data of each sample was 5.70 GB, and the GC content was 42.96–44.13%; Q30% was >92.22% (Appendix A). The overall distribution of the CK gene expression was lower than that of LCK, LHS, and LHT (Figure 6A). Principal component analysis (PCA) grouped the treatment repetitions (Figure 6C). The variabilities of PC1 and PC2 were 29.88% and 13.42%, respectively. The Pearson correlation coefficient (PCC) used to detect the correlation between samples (Figure 6B) indicated that the sampling measurements were reliable. 

Fold change > 1.5 and *p* < 0.05 were used as the screening criteria for differential genes. CK vs. LCK had 6418 DEGs, of which 3849 were upregulated and 2569 were downregulated. There were 81 DEGs between LCK and LHS, with 37 upregulated and 44 downregulated genes. LCK vs. LHT contained 25 DEGs, of which 20 were upregulated and 5 were downregulated. LHS vs. LHT had 408 DEGs, of which 48 were upregulated and 360 were downregulated (Figure 6D). A Venn diagram was used to show the number of common and unique differential genes among the four treatment groups (Figure 6E). The increased gene expression was inhibited following H_2_S removal.

### 2.7. GO Functional Enrichment and KEGG Pathway Annotation Analysis of DEGs

To explore the potential functions of the DEGs, we annotated their GO classifications. In total, 3927 DEGs and 62 DEGs were annotated into 56 GO terms for CK vs. LCK and LCK vs. LHS (Appendix A), respectively. The first 20 significantly enriched GO items were analyzed. In CK vs. LCK, the upregulated DEGs in BP were mainly enriched in metabolic process (1040 DEGs), cellular process (986 DEGs), single-organism process (659 DEGs), biological regulation (472 DEGs), and response to stimulus (307 DEGs). The entries with significantly upregulated differential genes in MF were binding (1349 DEGs), catalytic activity (1174 DEGs), transporter activity (130 DEGs), nucleic acid binding transcription factor activity (125 DEGs), and molecular function regulator (31 DEGs). The upregulated differential genes in CC were significantly enriched in membrane (806 DEGs), membrane part (723 DEGs), cell (643 DEGs), cell part (643 DEGs), and organelle (499 DEGs) (Figure 7A). In LCK vs. LHS, upregulated differential genes were significantly enriched in BP reproduction, immune system process, behavior, metabolic process, and cellular process. Significantly enriched genes in MF were transcription factor activity, protein binding, nucleic acid binding transcription factor activity, catalytic activity, signal transducer activity, and structural molecule activity. Significantly enriched genes in CC were in the extracellular region, cell, nucleoid, membrane, and cell junction (Figure 7B and Appendix A). The analysis results show that the stimulation response, cell membrane, and various genes related to the metabolic and catalytic activities of plants were mainly activated after the low-temperature stress, and the expression of differential genes related to cell membrane, nuclear, and catalytic activities could be reduced by spraying exogenous H_2_S, indicating that H_2_S effectively protects the integrity of the cell membrane and normal metabolic activities.

We further compared the DEGs to the Kyoto Encyclopedia of Genes and Genomes (KEGG) database to determine the role of H_2_S-mediated hypothermia stress response genes in some specific pathways (Figure 7C,D and Appendix A). In CK vs. LCK, the most significant differential gene enrichment pathways were (1) plant–pathogen interaction (ko04626), with a total of 256 DEGs (217 upregulated and 39 downregulated); (2) MAPK-signaling pathway–plant (ko04016), with a total of 146 DEGs (126 upregulated and 20 downregulated); and (3) plant hormone signal transduction (ko04075), with 166 DEGs (138 upregulated and 28 downregulated). For LCK vs. LHS, pentose and glucuronate interconversion (ko00040) had only one upregulated gene and two downregulated genes. For LHT vs. LHS, protein processing in the endoplasmic reticulum (ko04141), carbon metabolism (ko01200), RNA transport (ko03013), and amino sugars were also significantly enriched, along with nucleotide sugar metabolism (ko00520) and ribosomes (ko03010). These annotations provide a basis for understanding the genes and biological pathways involved in the differential response to low temperatures.

### 2.8. Pathway Analysis of Significant Enrichment Based on KEGG

Based on the KEGG analysis results, we further analyzed the three significantly enriched pathways in LCK vs. LHS (Figure 8A–C). When induced by exogenous H_2_S in the plant–pathogen interaction pathway, the downregulation of the expression of the Prf (disease resistance protein) gene led to the hypersensitive response in the pepper seedlings being reduced. The upregulated expressions of the WRKY22 (WRKY transcription factor 22) and PTI6 (pathogenesis-related genes transcriptional activator PTI6) genes and the activation of the downstream pathway promoted the accumulation of plant antibiotics, production of miRNA, and induction of defense-related genes. In the MAPK-signaling pathway–plant and plant hormone signal transduction pathways, ERF1 (ethylene-responsive transcription factor 1) was activated to promote the synthesis of ethylene. At the same time, the upregulated expression of GIBBERELLIN (GA) receptor GID2 (Gibberellin-insensitive DWARF2) and DELLA promoted the growth of the pepper seedlings.

### 2.9. qRT–PCR Verification

Nine randomly selected DEGs were verified using qRT–PCR (Figure 9). The results were consistent (88.9%) with the transcriptome sequencing of DEGs under each treatment, indicating that the sequencing data were reliable.

## 3. Discussion

The growth and growth period of pepper are adversely affected by low temperatures. Photosynthesis determines all metabolic activities during the autotrophic period of crops and is the basis for the yield and quality of vegetable crops. The effects of low temperature on photosynthesis mainly include the occurrence of photoinhibition and the decrease in some enzyme activities in the Calvin cycle. Photosynthetic parameters (Pn, Ci, Gs, Tr) can directly reflect the strength of photosynthesis [25,26]. A Pn decrease mainly includes stomatal factors and non-stomatal factors. Stomatal factors generally decrease or do not change with both Gs and Ci, while non-stomatal factors generally increase Gs and decrease Ci. In this study, when the pepper experienced low temperatures for 0–24 h, Ci increased and Gs decreased, indicating that Pn decreased due to non-stomatal factors in the early stage of low-temperature stress. When the pepper experienced low temperatures for 24–48 h, Ci and Gs decreased, indicating that Pn decreased due to stomatal factors in the later stage of low-temperature stress (Figure 2B,C). In addition, low-temperature stress reduced the activities of RubisCO, FBAase, and SPBase, which are the key enzymes of the photosynthetic Calvin cycle (Figure 2E–G); the same results were found in a spinach study [27]. In this experiment, exogenous H_2_S treatment significantly increased the light parameters and improved photosynthesis. At the same time, H_2_S increased the activity of key enzymes in the Calvin cycle and enhanced the capacity to capture light, which could potentially aid pepper seedlings under low-temperature stress. However, the H_2_S scavenger HT did the opposite, weakening the photosynthetic parameters and reducing the enzyme activity. Studies showed that H_2_S can significantly improve the photosynthetic gas exchange parameters of rape under Pb stress [28]. The possible reason behind this is that H_2_S induces stomatal closure and participates in the abscisic acid (ABA)-dependent signaling pathway by regulating ATP-binding cassette (ABC) transporters in guard cells [29]. Hence, the addition of exogenous H_2_S can regulate the ABA-induced H_2_O_2_ mediated stomatal closure and improve the intercellular CO_2_. 

Impaired photosynthesis leads to an imbalance in light–dark reactions, and the electrons released by the photosynthetic electron transport chain are received by oxygen to form O_2_^−^ [6]. At the same time, along with the lipid peroxidation of cell membranes, electrolyte leakage, the decrease in conductivity, the damage of the enzymatic oxidation system and ASA–GSH cycle, and MDA and ROS are also important indicators of lipid peroxidation of cell membranes and the first line of defense against low-temperature stress in plants [30]. In this study, after low-temperature stress, the pepper leaves rapidly curled and wilted (Figure 1A); with accelerated membrane lipid peroxidation (increased REC and MDA contents), the osmotic substance (Pro) content decreased, there was excessive accumulation of ROS (H_2_O_2_ and O_2_^·−^), and there was reduced oxidative stress (SOD, CAT, and POD). H_2_S could maintain the integrity of cell membranes, prevent membrane lipid peroxidation (reducing REC and MDA content), and increase the proline content and oxidative stress (Figure 1B,C), which were consistent with the results of Tang et al. [21], who found that low-temperature stress can cause ROS accumulation and increase MDA content.

Studies showed that low-temperature stress induces a rapid and transient increase in endogenous H_2_S in pepper, grape, and Arabidopsis [31,32]. In this study, low-temperature stress increased the content of endogenous hydrogen sulfide in pepper seedlings and increased the activities of LCD and DCD enzymes in the H_2_S synthesis pathway. Exogenous H_2_S treatment significantly enhanced the content of endogenous H_2_S and the enzyme activity of L/D-Cdes (Figure 5), and the same conclusion was also reached in cucumber by predecessors [16], indicating that exogenous H_2_S spray could improve the activity of key enzymes in the H_2_S synthesis pathway of pepper seedlings, thereby increasing the content of endogenous H_2_S and strengthening the tolerance of pepper seedlings to low temperatures. The exogenous H_2_S treatment improved the stability of the plasma membrane, decreased the levels of O_2_^·−^ and MDA, and increased the antioxidant activity (CAT, POD, glutathione reductase (GR), MDHAR, DHAR, and GPX) to remove overproduced ROS. This improved the frost resistance of the plants [33]. The ASA–GSH cycle is another antioxidant pathway in plants that alleviates oxidative damage by clearing H_2_O_2_ [34]. ASA, DHA, GSH, and GSSG are the main components of this cycle, and APX, DHAR, and MDHAR are the key enzymes of this cycle [35]. In this study, low-temperature stress for 24 h significantly decreased the contents of ASA and DHA and increased the contents of GSH and GSSG (Figure 4). With the extension of the low-temperature stress time, ASA/DHA gradually decreased, GSH/GSSG first increased then decreased, and MDHAR activity significantly decreased. The exogenous H_2_S decreased ASA/DHA, significantly increased GSSG content, increased the GSH/GSSG ratio, and increased the enzyme activities of APX, DHAR, and MDHAR (Figure 3). High GSH/GSSG and/or ASA/DHA ratios may be keys to effectively improving abiotic stress-induced ROS accumulation [36]. APX, which is an enzyme essential for detoxifying H_2_O_2_, has a strong affinity for H_2_O_2_ and can effectively remove ROS, even at low concentrations. MDHAR catalyzes the reduction of MDHA to ASA and MDHA, while NADPH catalyzes the reduction of GSSG to GSH in the presence of GR to generate DHA from MDHA and uses ASA as an electron donor to convert H_2_O_2_ into H_2_O [37]. In addition, in this study, compared with the exogenous H_2_S treatment, HT aggravated the ROS imbalance by reducing the endogenous H_2_S content, partially eliminating the beneficial effects of H_2_S on the oxidative stress and antioxidant defense system, which indicates that the H_2_S could effectively alleviate the damage of low temperature on the pepper seedlings. Similar conclusions were also obtained for waterlogging damage of peach seedlings [38] and boron stress of tomato [39].

To comprehensively study the molecular mechanism of the H_2_S alleviating effect on pepper seedlings under low-temperature stress, we constructed 12 cDNA libraries using the Illumina sequencing platform. In this study, RNA samples of pepper seedlings were analyzed to explore the changes in the biological metabolic pathways related to H_2_S alleviation in response to low-temperature stress. Illumina sequencing identified 6418 DEGs in CK vs. LCK, 7595 in CK vs. LHS, 6367 in CK vs. LHT (Appendix A), and 81 in LCK vs. LHS (Figure 6). The results show that H_2_S induced more DEGs, which is consistent with the changes in plant phenotype, membrane lipid peroxidation, and oxidative stress after low-temperature stress (Figure 1 and Figure 3). Similar results were observed in cucumbers, where the overexpression of the *Gs WRK* transcription factor resulted in high Pro accumulation, low malondialdehyde levels, and greater cold resistance [40]. 

Plants have evolved strategies to cope with cold stress by coordinating cold and hormone-signaling pathways [41]. The mitogen-activated protein kinase (MAPK) cascade was suggested to play an important role in the low-temperature response, and protein kinase is a connection point for rapid osmotic regulation and signal transduction under salt stress [42]. MAPKs also confer salt tolerance in plants and induce defense responses. For example, zmMPK5, zmMAPK3, and zmMPK6 improve salt tolerance in tobacco [43] and rice [44]. In this study, the two pathways by which LCK vs. LHS were significantly enriched were MAPK-signaling pathway–plant (ko04016) and plant hormone signal transduction (ko04075) (Figure 7C,D). Exogenous H_2_S upregulated the expression of transcription factors WRKY22 and PTI6; induced defense genes; and activated the ethylene and gibberellin synthesis receptors ERF1, GDI2, and DELLA, improving the resistance to low-temperature chilling injury of pepper seedlings (Figure 8A–C). Several stress stimuli and growth signals induce the production of endogenous H_2_S via enzymatic pathways. For example, exogenous H_2_S activates DES1 through per-sulfidation [45], while TGA3 promotes an increase in LCD transcription levels [46]. Drought-induced hormones (ABA, jasmonic acid (JA), and ethylene) and ROS signals differ between different plants, promoting the accumulation of H_2_S in guard cells and initiating signals downstream of H_2_S to induce stomatal closure [47,48]. The results show that exogenous H_2_S promoted the connection of the MAPK signal pathway and plant hormone signal transduction, activated the expression of defense genes, and enhanced the resistance to low-temperature chilling injury of pepper seedlings. In addition, the plant–pathogen interaction (ko04626) was also significantly enriched, suggesting that exogenous H_2_S also promoted the expression of genes related to the plant–pathogen interaction. In this study, we mainly focused on physiological changes, specific pathways, and transcription factors. In future studies, we will further investigate the application of exogenous H_2_S and the biogenic analysis of specific genes and transcription factor families.

## 4. Materials and Methods

### 4.1. Experimental Material 

The experiment was conducted in March 2022 at the College of Horticulture, Sichuan Agricultural University (Chengdu, China). Seeds of ‘Long Yun2’ pepper were stored at and obtained from the vegetable laboratory of the Horticulture College, Sichuan Agricultural University. The H_2_S donor, namely, sodium hydrosulfide (NaHS), was purchased from McLean (Shanghai, China), and the H_2_S scavenger, namely, hypotaurine (HT), was purchased from Sigma (Shanghai, China).

### 4.2. Experimental Design 

Selected whole pepper seeds were soaked in warm water at 55 °C for 15 min and distilled water at 25 °C for 10 h. The seeds were placed in an artificial climate box (Model: RGX300EF, Tianjin Test Instrument Co., Ltd., Tianjin, China) at 25/18 °C and 12/12 h for germination.

When 80% of the seeds were germinated, the seeds were sown in a 12 × 6 (3 × 3 ×5 cm) seedling tray (seedling medium: peat soil: vermiculite: composting = 5:3:2) with a seedling environment temperature of 25/18 °C, photoperiod of 12/12 h (day and night), light intensity of 300 µmol m^−2^ s^−1^, and relative humidity of 75%. In the process of seedling cultivation, the plants were watered every 7 days so that the soil water content reached ~70%. When the seedlings grew to 3–4 true leaves, they were transplanted into a pot of 10 × 15 cm. When the seedlings grew to 7 to 8 true leaves, the seedlings were treated with exogenous deionized water, H_2_S, and HT for 4 consecutive days (200 mL/day). On day 5, the seedlings were placed in an artificial climate chamber for low-temperature stress (5 ± 1 °C). The photoperiod was 12/12 h (day and night), the light intensity was 300 µmol m^−2^ s^−1^, and the relative humidity was 75%. 

Four treatments were set up in the experiment: (1) normal temperature control (CK), deionized water, 25 °C; (2) low-temperature control (LCK), deionized water, 5 °C; (3) low-temperature H_2_S treatment (LHS), 0.5 mmol L^−1^ H_2_S, 5 °C; and (4) low-temperature HT treatment (LHT), 0.2 mmol L^−1^ HT, 5 °C. All processing light intensities were 300 µmol m^−2^ s^−1^. Each treatment included 6 seedlings and was repeated three times. Leaf samples were collected at different time points of the low-temperature stress (0, 6, 12, 24, and 48 h) for the determination of physiological indices. All samples were biologically replicated three times. Samples were immediately frozen in liquid nitrogen and stored at −80 °C.

### 4.3. Determination of Physiological and Biochemical Indices

#### 4.3.1. MDA and Relative Electrolyte Leakage (REL)

The MDA content was slightly modified [49]. In brief, 0.5 g of fresh leaf samples were homogenized at 4 °C with 10% (*w*/*v*) trichloroacetic acid (TCA) and centrifuged at 3000× *g* for 10 min. The supernatant was incubated with an equal volume of 0.5% thiobarbituric acid (TBA) at 100 °C for 30 min and the absorbance was measured at 450, 532, and 600 nm.

A 0.1 g blade punched with a 0.5 cm diameter punch was weighed, placed into a 50 mL centrifuge tube, mixed with 30 mL deionized water, and placed on a shaker at 150 rpm for 6 h. The conductivities of deionized water (S_0_) and the immersion solution (S_1_) were measured using a conductance meter. The immersed solution was boiled in a water bath for 30 min and cooled to room temperature before measuring the conductivity of the solution (S_2_). The relative electrolyte leakage was calculated as follows [50]: REL (%) = (S_1_ − S_0_) × 100/(S_2_ − S_0_)

#### 4.3.2. Photosynthetic Indices Assay

The photosynthetic parameters were determined using a Li-Cor 6400XT (Gene Company Limited, Hong Kong, China) portable photosynthesis measurement system. Photosynthetic indices were measured using fully unfolded leaves. With a red/blue LED as the light source, the flow rate was 500 mL min^−1^, the CO_2_ concentration was 400 μmol^−1^, and the PAR was 1000 μmol m^−2^ s^−1^. The photosynthetic rate (Pn), stomatal conductance (Gs), transpiration rate (Tr), and intercellular CO_2_ concentration (Ci) of the best functional leaves (3rd to 4th true leaves) of pepper seedlings in each treatment were determined, and each treatment was repeated three times. The enzyme activity changes of RubisCO, FBA, and SBP were determined using an ELISA kit (Shanghai Fwei Biotechnology Co., Ltd., Fengxian District, Shanghai, China).

#### 4.3.3. Determination of Pro, ROS, Antioxidant Enzymes, and ASA–GSH

The Pro content was determined using the sulfosalicylic acid method [51]. The O_2_^−^ content was determined using the p-aminobenzene-sulfonic acid method [52], and the H_2_O_2_ content was determined using potassium iodide spectrophotometry [53]. POD, SOD, and CAT activities were determined using the guaiac method, nitrogen blue tetrazole (NBT), and ultraviolet absorption, respectively [54]. The activities of APX, MDAHR, and DHAR were determined as previously described [55]. ASA, dehydroascorbate (DHA), GSH, and glutathione disulfide (GSSG) contents were determined as described by Noctor and Foyer [56].

#### 4.3.4. Measurements of Endogenous Hydrogen Sulfide and Synthase

The determination of endogenous H_2_S was slightly modified from the methylene blue method described by Sekiya et al. [57]: 0.1 g fresh functional leaves were added into 0.9 mL 20 mM pre-cooled Tris-HCl buffer (pH 8.0), ground into a homogenate, and centrifuged at 4 °C and 12,000× *g* for 20 min; the supernatant was extracted for further analysis. The absorption wells containing zinc acetate were placed in a small test tube containing the supernatant. After adding 100 μL of 30 mM FeCl_3_ (dissolved in 1.2 M HCl) and 100 μL of 20 mM N, n-dimethyl-p-phenylenediamine (dissolved in 7.2 M HCl), the test tube was sealed with a sealing film and allowed to react at 37 °C for 30 min. The absorbance was measured at 670 nm.

The activity of L-/D-cysteine dehydrase was determined by measuring the H_2_S released by L-/D-cysteine (containing DTT), as previously described by Riemenschneider et al. [58]. The 1 mL reaction system comprised 0.8 mM L-cysteine, 2.5 mM DTT, 100 mM Tris-HCl (pH 9.0), and enzyme extract. After adding the L-cysteine, the reaction was incubated at 37 °C for 30 min. Finally, 100 μL of 30 mM FeCl_3_ (dissolved in 1.2 M HCl) and 100 µL of 20 mM N were added. N-dimethyl-p-phenylenediamine (dissolved in 7.2 M HCl) was added to terminate the reaction. The absorbance was measured at 670 nm. The procedure for the determination of D-cysteine de mercapto activity was the same as that for L-cysteine de mercapto activity, except that L-cysteine was replaced with D-cysteine and the Tris-HCl had a pH value of 8.0.

### 4.4. RNA Extraction, Sample Detection, Library Construction and Sequencing, and RNA-seq Data Analysis

Total RNA was extracted from 12 pepper leaf samples (four treatments and three replicates) subjected to 24 h of low-temperature stress. After the pepper leaf samples were ground into a powder with liquid nitrogen, total RNA was extracted using a TRIzol reagent according to the manufacturer’s instructions (Invitrogen, Carlsbad, CA, USA). The purity and concentration of the total RNA were detected using a NanoDrop 2000 spectrophotometer, and the integrity of the RNA was detected using an Agient2100/LabChip GX (Qingpu District, Shanghai, China). 

Following sample qualification, the library was constructed, the eukaryotic mRNA was enriched with magnetic beads with Oligo (dT), and the mRNA was randomly interrupted using a fragmentation buffer. The first and second cDNA strands were synthesized using the mRNA as a template. The purified double-stranded cDNA was end-repaired, the tail was added, and the sequencing joints were connected. AMPure XP beads were used for fragment size selection, and a cDNA library was obtained using PCR enrichment. Finally, Beijing Biomarker Technologies Co, Ltd. (http://www.biomarker.com.cn/, accessed on 26 August 2022 Beijing, China), was commissioned to perform high-throughput sequencing based on Illumina; the cDNA library was sequenced using the NovaSeq6000 (Accessed on 26 August 2022) sequencing platform and raw data were obtained. Data were analyzed using the bioinformatics analysis process provided by BMK Cloud (www.biocloud.net. Accessed on 26 August 2022). High-quality clean reads were obtained by filtering raw data to remove reads containing connectors and low-quality reads. The obtained clean reads were compared with the pepper reference genome using the HISAT 2.0 software (Capsicum_annuum.GCF_000710875.1. genome. fa). Rapid sequence alignment and subsequent analysis were conducted, and, after obtaining the mapped data, library quality assessment, differential expression analysis, gene function annotation, and function enrichment for each sample were also performed. The number of gene-based reads was compared with the count in each sample. The DESeq3.0 software was used to screen the DEGs, with fold change ≥ 1.5 and FDR < 0.05 as the screening criteria.

### 4.5. Identification and Analysis of DEGs

Fragments per kilobase of coding sequence per million reads (FPKM) were used to quantify the mapped gene expression levels. Differential expression analysis was performed using DESeq2 software. The thresholds of FDR < 0.01 and |log2 (fold change)| ≥ 1 were used to identify differentially expressed genes (DEGs). Gene Ontology (GO) enrichment analysis of the DEGs was performed using the GoSeq R package [59]. The KOBAS (Accessed on 12 November 2022.) software was used to test the statistical enrichment of the DEGs in the Kyoto Encyclopedia of Genes and Genomes (KEGG) pathway [60].

### 4.6. qRT–PCR

qRT–PCR was performed to verify the reliability of the RNA-seq data. Specific primers were designed using the website (http://primer3.ut.ee/ Accessed on 15 November 2022) according to design principles (Appendix A). Total RNA was isolated from the leaf tissue using the RNA simple Total RNA kit (Tiangen, Changping District, Beijing, China). cDNA was synthesized from total RNA using the HiScript II Q Select RT SuperMix qPCR kit (Vazyme, Haidian District, Beijing,China). qRT–PCR was performed on the LightCycler 480II (Roche, Changping District, Beijing, China) using AceQ^®^ qPCR SYBR^®^ Green Master Mix (Vazyme, China). The PCR reaction conditions were as follows: denaturation at 95 °C for 5 min and then 40 cycles of 95 °C for 10 s and 60 °C for 30 s. The expression level was calculated using the 2^−ΔΔCt^ method [61]. Three independent experimental replicates were analyzed for each sample. 

### 4.7. Statistical Analysis

Statistical tests were performed using the SPSS software (version 26.0). Differences between treatments were determined using the least significant difference (Duncan’s) test, and the threshold for statistical significance was *p* < 0.05. Data are presented as the mean ± SD of three biological replicates.

## 5. Conclusions

In this study, the effects of exogenous H_2_S on pepper seedlings after low-temperature stress were analyzed. In terms of affecting physiological characteristics to alleviate chilling injury symptoms, exogenous H_2_S could increase the content of endogenous H_2_S and its synthase activity, enhance the antioxidant capacity of membrane lipids, and protect the integrity of the membrane system. In addition, exogenous H_2_S could promote the Cartesian cycle, protect the integrity of photosynthetic organs, improve the activity of enzyme systems and ASA–GSH oxidase, and decompose excess ROS, thereby enhancing tolerance to low temperatures in pepper seedlings. In addition, the H_2_S scavenger hypotaurine aggravated the ROS imbalance by reducing the endogenous H_2_S content, which partially eliminated the beneficial effects of H2S on oxidative stress and the antioxidant defense system, indicating that H_2_S could effectively alleviate the damage of low temperature on pepper seedlings. H_2_S could induce MAPK-signaling pathway–plant and plant hormone signal transduction; upregulate the expression of transcription factors WRKY22 and PTI6; induce defense genes; and activate the ethylene and gibberellin synthesis receptors ERF1, GDI2, and DELLA. The plant–pathogen interaction was also significantly enriched, suggesting that exogenous H_2_S also promoted the expression of genes related to plant–pathogen interaction. The results of this study provide novel insights into the molecular mechanisms and genetic modifications of H_2_S that mitigate the hypothermic response.

## Figures and Tables

**Figure 1 ijms-24-16337-f001:**
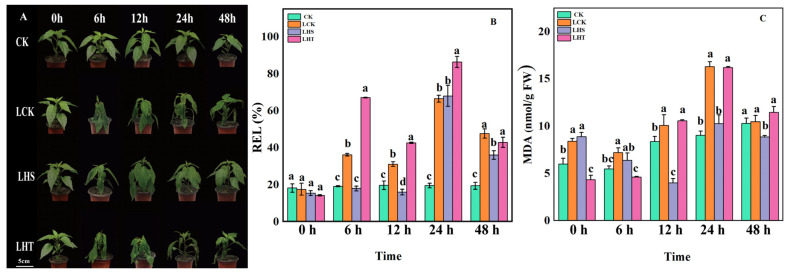
Changes in seedling phenotype and membrane system after low-temperature stress. (**A**) Phenotype; (**B**) relative electrolyte leakage (REL); (**C**) malondialdehyde (MDA). Each data point represents the mean of three repeated samples, and the result is the mean ± standard deviation. According to Duncan’s test, different letters indicate a statistically significant difference (*p* < 0.05).

**Figure 2 ijms-24-16337-f002:**
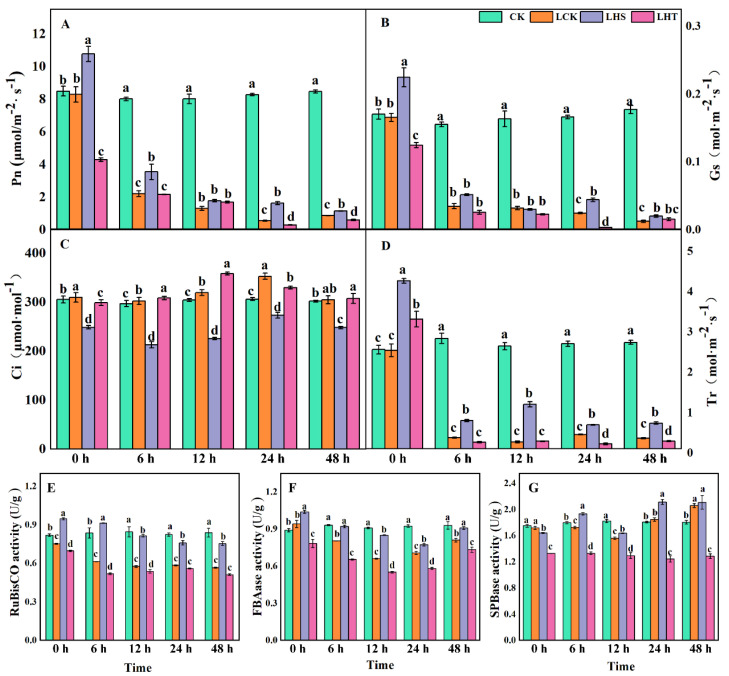
Effects of low-temperature stress on photosynthetic parameters and photosynthetic pigments of pepper seedlings. (**A**) Photosynthetic rate (Pn); (**B**) stomatal conductance (Gs); (**C**) intercellular CO_2_ concentration (Ci); (**D**) transpiration rate (Tr); (**E**) Ribulose diphosphate carboxylase/oxidase (RubisCO); (**F**) Fructose-1; 6-diphosphate aldolase (FBAase); (**G**) sepullose-1; 7-diphosphatase (SBPase). Each data point represents the mean of three repeated samples, and the result is the mean ± standard deviation. According to Duncan’s test, different letters indicate a statistically significant difference (*p* < 0.05).

**Figure 3 ijms-24-16337-f003:**
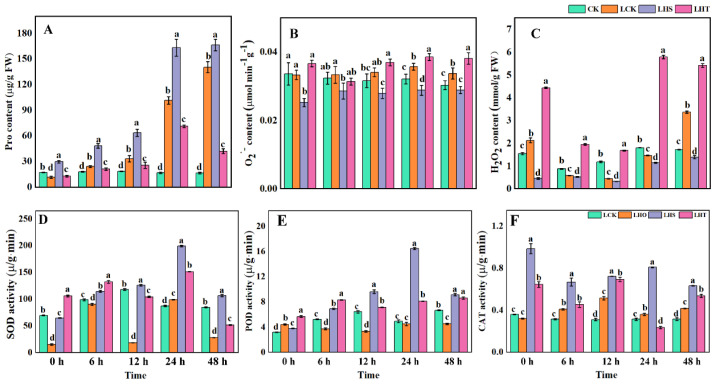
Effects of low-emperature stress on ROS in an enzymatic oxidation system. (**A**) Proline (Pro); (**B**) superoxide anion (O_2_^−^); (**C**) hydrogen peroxide (H_2_O_2)_; (**D**) superoxide dismutase (SOD); (**E**) peroxidase (POD); (**F**) catalase (CAT). Each data point represents the mean of the repeated samples, and the result is the mean ± standard deviation. According to Duncan’s test, different letters indicate a statistically significant difference (*p* < 0.05).

**Figure 4 ijms-24-16337-f004:**
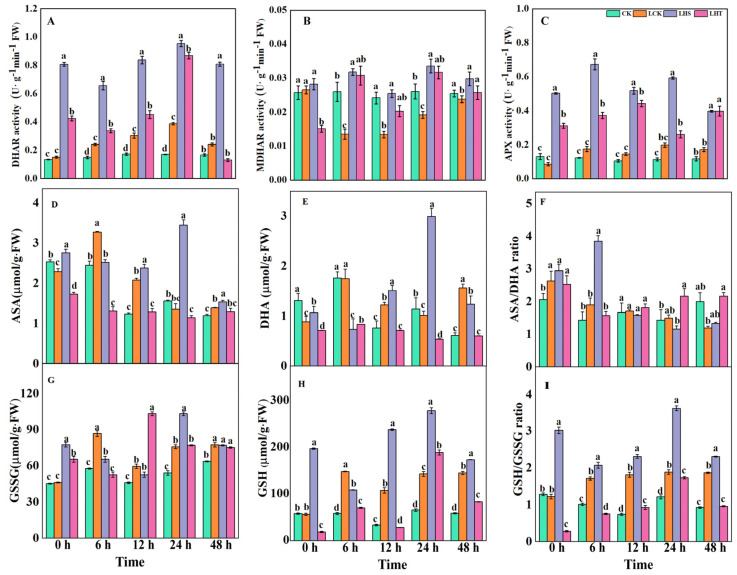
Effects of low-temperature stress on the ASA–GSH cycle. (**A**) Dehydroascorbate reductase (DHAR); (**B**) monodehydroascorbate reductase (MDHAR); (**C**) ascorbate peroxidase (APX); (**D**) ascorbic acid (ASA); (**E**) dehydroascorbate (DHA); (**F**) ASA/DHA; (**G**) glutathione disulfide (GSSG); (**H**) glutathione (GSH); (**I**) GSH/GSSG. Each data point represents the mean of three repeated samples, and the result is the mean ± standard deviation. According to Duncan’s test, different letters indicate a statistically significant difference (*p* < 0.05).

**Figure 5 ijms-24-16337-f005:**
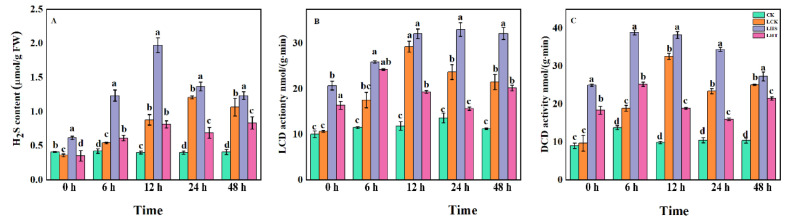
Effects of low-temperature stress on endogenous H_2_S and key synthase in pepper seedlings. (**A**) Endogenous H_2_S; (**B**) L-cysteine desulfhydrase (LCD); (**C**) D-Cysteine desulfhydrase (DCD). Each data point represents the mean of three repeated samples, and the result is the mean ± standard deviation. According to Duncan’s test, different letters indicate a statistically significant difference (*p* < 0.05).

**Figure 6 ijms-24-16337-f006:**
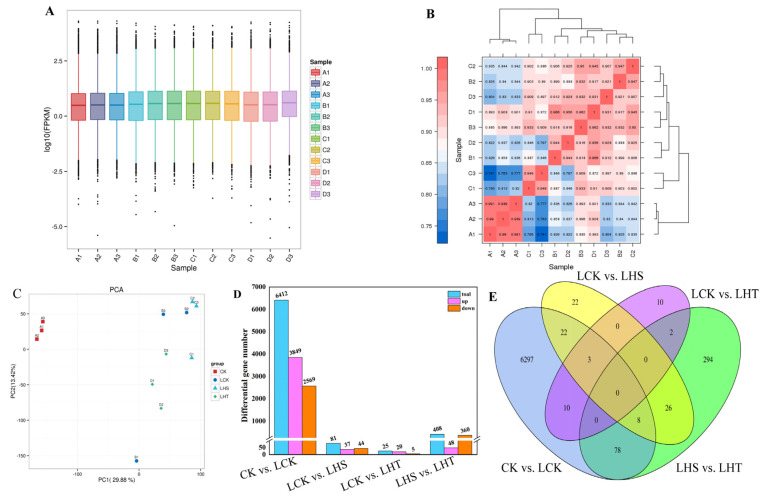
Gene expression and differential genes of intertreatment samples. (**A**) All FPKM box plot; (**B**) sample cluster; (**C**) principal component analysis (PCA); (**D**) numbers of differential genes (DEGs); (**E**) intertreatment Venn diagram.

**Figure 7 ijms-24-16337-f007:**
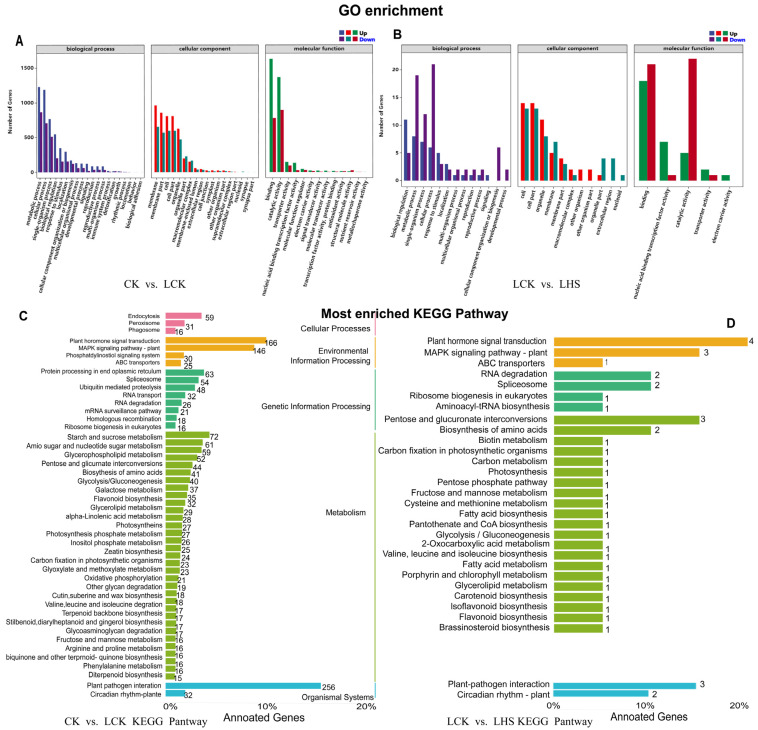
GO enrichment and KEGG pathway enrichment analysis of DEGs. (**A**) Significantly enriched GO terms in CK vs. LCK; (**B**) significantly enriched GO terms in LCK vs. LHS; (**C**) statistical analysis of annotated unique genes in KEGG pathways in CK vs. LCK; (**D**) statistical analysis of annotated unique genes in KEGG pathways in LCK vs. LHS.

**Figure 8 ijms-24-16337-f008:**
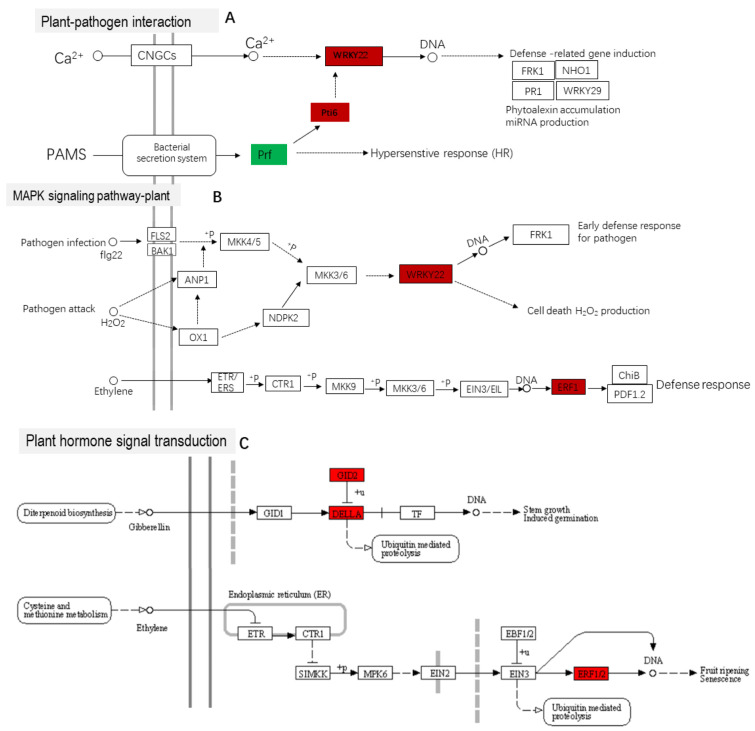
Pathway analysis of significant enrichment of KEGG. (**A**) Plant–pathogen interaction; (**B**) MAPK-signaling pathway–plant; (**C**) plant hormone signal transduction. Upregulated genes are marked by red boxes, downregulated genes are marked by green boxes, and white represents no significant change.

**Figure 9 ijms-24-16337-f009:**
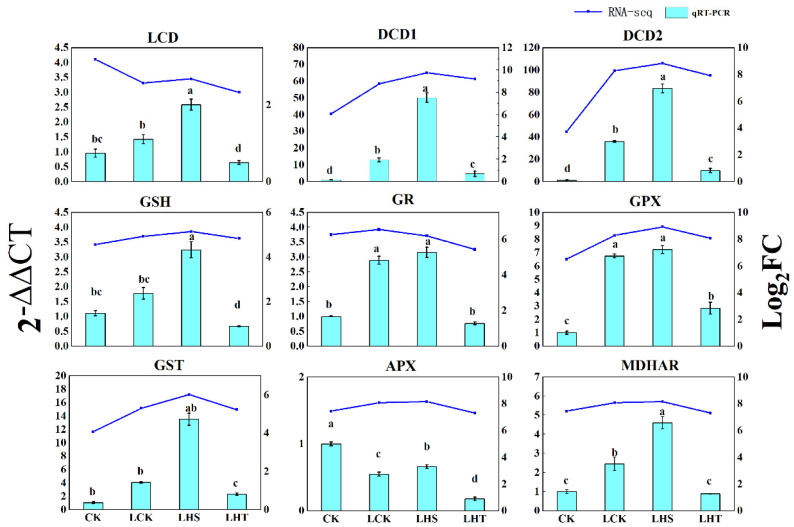
Expression levels of randomly selected DEGs. The column diagram and line chart represent data from qRT−PCR and RNA-seq, respectively. According to Duncan’s test, different letters indicate a statistically significant difference (*p* < 0.05).

## Data Availability

Data are contained within the article.

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
