# Peer review of "Molecular Regulatory Mechanism of Exogenous Hydrogen Sulfide in Alleviating Low-Temperature Stress in Pepper Seedlings"

_ijms, 2023, doi:10.3390/ijms242216337_

Round 1

Reviewer 1 Report

Comments and Suggestions for Authors

This manuscript provides substantial data on molecular aspects shown to regulate metabolism and gene action associated with chilling injury in pepper seedlings. The temperature and duration treatments used are specific to low temperature chilling injury. The term cold hardiness is a broad term associated with a broad range of low temperature damage including sub freezing temperatures. It is more precise and preferable to refer to the conditions used in this study as chilling injury.

The report provides an abundance of data that convincingly tie together many known and reported aspects of low temperature, chilling injury.

The following edits are recommended.

L19: improve resistance to low temperature chilling injury in 'Long Yun2' pepper seedlings.

L23: suggest, potentially enhancing resistance to low temperature chilling injury. Because no growth studies were done  there is no evidence to refer to promoting growth.

L30: impacting the productivity...

L85: the Capsicum

L91: cultivars..varieties

L95: Seeds of 'Long Yun2' pepper were stored and obtained from the vegetable....

L105: germinated, they seeds they

L106: composting

L512: In terms of alleviating chilling injury symptoms physiological characteristics...

L521: Because no growth studies were done remove "promoting the growth of pepper seedlings...

No follow-up studies were done, thus enhancing cold tolerance should be modified.

This work should be published. One downside is because of the extensive use of acronyms, necessitating constant reference to the list of acronyms, while necessary, makes the paper laborious to read and evaluate. When appropriate, including a definition of the acronym especially in the data tables when first introduced, may make the paper more readable.

Author Response

November 11, 2023

Dear Editors and Reviewers:

Thank you for your letter and for the reviewers' comments concerning our manuscript entitled "Molecular regulatory mechanism of exogenous hydrogen sulfide in alleviating low-temperature stress in pepper seedlings "(ID: ijms-2692160). Those comments are all valuable and very helpful for revising and improving our paper, as well as the important guiding significance tour researches. We have studied comments carefully and have made correction which we hope meet with approval. Revised portion are marked in red in the paper. The main corrections in the paper and the responds to the reviewer's comments are as flowing:

Reviewer:

-1. L19: improve resistance to low temperature chilling injury in 'Long Yun2' pepper seedlings.

-Response: Thank you very much for the reviewer's suggestion. L23: suggest, potentially enhancing resistance to low temperature chilling injury. Because no growth studies were done there is no evidence to refer to promoting growth.

-Response: We have completed the modification according to the suggestion and marked L25 and L32 in red font in the paper.

-2. mpacting the productivity...; L85: the Capsicum; L91: cultivars. .varieties; L95: Seeds of 'Long Yun2' pepper were stored and obtained from the vegetable....; L105: germinated, they seeds they; L106: composting

-Response: Thank you very much for the reviewer's suggestion. We have completed the modification according to the suggestion and marked it in red font

-3. In terms of alleviating chilling injury symptoms physiological characteristics...

L521: Because no growth studies were done remove "promoting the growth of pepper seedlings...

-Response: Thank you very much for the reviewer's valuable comments. We have modified L416 in red font in the conclusion of the manuscript, and the "promoting the growth of pepper seedlings..." Cut out.

We tried our best to improve the manuscript and made some changes in the manuscript. These changes will not influence the content and framework of the paper. And here we did not list the changes but marked in red in revised paper. We appreciate for Editors/Reviewers’ warm work earnestly, and hope that the correction will meet with approval.

Once again, thank you very much for your comments and suggestions.

Sincerely,

Song Xueping

College of Horticulture, Sichuan Agricultural University, Chengdu, Sichuan, China

Email address: [email protected]

Corresponding author: [email protected]

Reviewer 2 Report

Comments and Suggestions for Authors

The manuscript titled “Molecular regulatory mechanism of exogenous hydrogen sulfide in alleviating low-temperature stress in pepper seedlings” focuses on the protective effect of exogenous H2S application on pepper cold tolerance. The manuscript is reasonably well-written and its results, although not entirely novel, can be useful both for agronomy and for basic plant sciences. There are only a few issues that I consider should be addressed before being suitable for publication:

The results from the LHT treatment (cold + the H2S scavenger hypotaurine) are not well described. This treatment could provide useful information about which parts of the stress response require H2S to properly function. It should be more featured both in the “results” and, particularly, in the “discussion” sections. Additionally, authors should provide the appropriate citations that describe the H2S-scavenging properties of hypotaurine.

Some of the sentences found in the results sections are not completely clear. Authors sometime describe the increase or decrease of certain parameter over time, but they do not clearly explain what they are comparing it to. The reader may assume that the data is compared against controls, but this should be explicitly stated. For example, in Line 270 authors write that “With the extension of low-temperature stress, Pro content continued to increase, O2·- content first increased and then decreased, and H2O2 content first decreased and then increased”. It can be assumed that they are describing the Low temperature treatment, but there are technically three treatments involving low-temperature stress, and it is not clear until later in the text which one they are specifically referring to. Additionally, “first increased and then decreased” are a bit too vague descriptions, considering that the authors have specific time-points to work with. Lastly, some of the described trends not always match what can be observed in the figures. In the above case, for example, it is stated that “O2·- content first increased and then decreased”, when Figure 3B never shows the O2·- levels of LCK treatment being higher than controls. Unless they are comparing it to something else, but that cannot be guessed from the text. It is important that authors focused on writing unambiguous sentences, so that the reader do not need to guess the meaning.

Some of the claims made in the discussion and conclusions are not completely supported by the data, or their formulation leads to incorrect assumptions:

·       L419: “the decrease in Pn may be linked to nonstomatal factors, including limited carbon assimilation and poor photosynthetic performance”. Pn, carbon assimilation and photosynthetic performance are basically synonyms of each other. Carbon assimilation and photosynthetic performance are not “nonstomatal factors”, they are the end product of the whole process, which includes both stomatal and nonstomatal factors. Authors make a similar mistake in Line 425, when claiming that “Pn limits the operation of the dark reaction and affects the accumulation of compounds”. The photosynthetic rate is the final product of the whole process and, as such, it is going to be determined, among many other factors, by the capacity of the plant to perform the dark reactions of the photosynthesis. The cause-effect goes the other way around to what the authors claim.

·       L427 “H2S also enhanced the light availability and efficiency of pepper seedlings under low-temperature stress by improving the Calvin cycle”. The light availability is an external factor that cannot be enhanced by H2S. The capacity to capture that light could, potentially, be enhanced by the supply pf H2S, but it is unlikely that it would be through the enhancement of the Calving cycle. If there are studies that prove it otherwise, authors should provide the appropriate citation.

·       In the section between Lines 429 and 434 authors describe how H2S has been shown to improve photosynthetic rates under Pb toxicity, and then link it to the improvement in Pn rates that they have observed in their study. These two stresses are very different from each other, and even if H2S can improve the Pn rates in both of them, that does not necessary mean that it is acting in the same way. If authors think that H2S is acting through the same molecular mechanisms in both stresses, then it should be better described which those molecular mechanisms are.

·       L449 “Exogenous H2S enhanced the Calvin cycle during photosynthesis, and the enzyme activities of endogenous H2S and D/L-Cdes” The way the sentence is formulated, it makes it look like H2S content is an enzyme activity, when in reality only D- and L-Cdes are the enzymes. This should be rephrased.

·       L451 “Low-temperature stress induces a rapid and transient increase in endogenous H2S in pepper, grape, and Arabidopsis (Du et al., 2017; Fu et al., 2013), which indicates that exogenous H2S not only increases the content of endogenous H2S and cysteine but also plays an important role in protecting the cell membrane and biological metabolic processes, further amplifying the physiological effects of H2S”. The fact that endogenous H2S increased under low temperature stress does not necessary mean that exogenous H2S will increase endogenous H2S and cysteine content, nor protect the cell membrane or metabolic processes. To be clear, I am not claiming that it would not lead to those effects, particularly since the authors have the data that backs it up, but this sentence does not provide the information that would lead to that conclusion on its own. This should be rephrased to avoid logical mistakes.

·       L518. H2S does not induce “plant-pathogen interactions”, since there are no pathogens in this study. It does however induce the expression of genes related to plant-pathogen interactions. This should be rephrased.

·       L521. I do not think that the growth of pepper plants was quantified in this study. If that the case, this should be reformulated ta make it clear that this was not an observation, but an educated guess.

These are a couple of other minor comments:

·       L81. Range of

·       L125 Isn’t “relative conductivity” much more often referred to as “electrolytic leakage”?

·       Statistics should be explained in each figure description. What exactly is being compared to what.

·       In Figure 6E the ends of two of the columns occur in the middle of the “axis break”, and are therefore not visible.

Comments on the Quality of English Language

The English language of the manuscript is adequate in general, and only few minor mistakes/typos should be corrected. However, as explained above, some sentence should be rephrased for the sake of clarity and avoidance of logical mistakes.

Author Response

November 11, 2023

Dear Editors and Reviewers:

Thank you for your letter and for the reviewers' comments concerning our manuscript entitled "Molecular regulatory mechanism of exogenous hydrogen sulfide in alleviating low-temperature stress in pepper seedlings "(ID: ijms-2692160). Those comments are all valuable and very helpful for revising and improving our paper, as well as the important guiding significance tour researches. We have studied comments carefully and have made correction which we hope meet with approval. Revised portion are marked in red in the paper. The main corrections in the paper and the responds to the reviewer's comments are as flowing:

Reviewer:

-1. The results from the LHT treatment (cold + the H2S scavenger hypotaurine) are not well described. This treatment could provide useful information about which parts of the stress response require H2S to properly function. It should be more featured both in the “results” and, particularly, in the “discussion” sections. Additionally, authors should provide the appropriate citations that describe the H2S-scavenging properties of hypotaurine.

-Response: According to the reviewer's comments, we re-checked and supplemented the description of LHT processing in the results. In the discussion L372-L377, we also added relevant descriptions and references and marked them with red font.

-2. Some of the sentences found in the results sections are not completely clear. Authors sometime describe the increase or decrease of certain parameter over time, but they do not clearly explain what they are comparing it to. The reader may assume that the data is compared against controls, but this should be explicitly stated. For example, in Line 270 authors write that “With the extension of low-temperature stress, Pro content continued to increase, O2·- content first increased and then decreased, and H2O2 content first decreased and then increased”. It can be assumed that they are describing the Low temperature treatment, but there are technically three treatments involving low-temperature stress, and it is not clear until later in the text which one they are specifically referring to. Additionally, “first increased and then decreased” are a bit too vague descriptions, considering that the authors have specific time-points to work with. Lastly, some of the described trends not always match what can be observed in the figures. In the above case, for example, it is stated that “O2·- content first increased and then decreased”, when Figure 3B never shows the O2·- levels of LCK treatment being higher than controls. Unless they are comparing it to something else, but that cannot be guessed from the text. It is important that authors focused on writing unambiguous sentences, so that the reader do not need to guess the meaning.

-Response: Thank you very much for the reviewer's comments. We have carefully checked the manuscript, expressed the comparison between processing in clearer language, and marked the changes in red font。

-3. L419: “the decrease in Pn may be linked to nonstomatal factors, including limited carbon assimilation and poor photosynthetic performance”. Pn, carbon assimilation and photosynthetic performance are basically synonyms of each other. Carbon assimilation and photosynthetic performance are not “nonstomatal factors”, they are the end product of the whole process, which includes both stomatal and nonstomatal factors. Authors make a similar mistake in Line 425, when claiming that “Pn limits the operation of the dark reaction and affects the accumulation of compounds”. The photosynthetic rate is the final product of the whole process and, as such, it is going to be determined, among many other factors, by the capacity of the plant to perform the dark reactions of the photosynthesis. The cause-effect goes the other way around to what the authors claim.

-Response: Thank you very much for the reviewer's comments. We have carefully checked the manuscript and revised the arguments on photosynthetic parameters (L308-L317), and marked the changes with red font.

-4. L427 “H2S also enhanced the light availability and efficiency of pepper seedlings under low-temperature stress by improving the Calvin cycle”. The light availability is an external factor that cannot be enhanced by H2S. The capacity to capture that light could, potentially, be enhanced by the supply pf H2S, but it is unlikely that it would be through the enhancement of the Calving cycle. If there are studies that prove it otherwise, authors should provide the appropriate citation. In the section between Lines 429 and 434 authors describe how H2S has been shown to improve photosynthetic rates under Pb toxicity, and then link it to the improvement in Pn rates that they have observed in their study. These two stresses are very different from each other, and even if H2S can improve the Pn rates in both of them, that does not necessary mean that it is acting in the same way. If authors think that H2S is acting through the same molecular mechanisms in both stresses, then it should be better described which those molecular mechanisms are.

-Response: Thanks very much for the reviewer's comments. After careful examination of the language in the manuscript, we modified some of the previous arguments with inappropriate expressions, and described the molecular mechanism of H2S to alleviate low temperature and lead stress with new references. In the manuscript, L319-L329 marked the modification in red font.

-5. L449 “Exogenous H2S enhanced the Calvin cycle during photosynthesis, and the enzyme activities of endogenous H2S and D/L-Cdes” The way the sentence is formulated, it makes it look like H2S content is an enzyme activity, when in reality only D- and L-Cdes are the enzymes. This should be rephrased.

-Response: Thanks very much for the reviewer's valuable comments, we reorganized the language and marked it in red font in the manuscript.

-6: L451 “Low-temperature stress induces a rapid and transient increase in endogenous H2S in pepper, grape, and Arabidopsis (Du et al., 2017; Fu et al., 2013), which indicates that exogenous H2S not only increases the content of endogenous H2S and cysteine but also plays an important role in protecting the cell membrane and biological metabolic processes, further amplifying the physiological effects of H2S”. The fact that endogenous H2S increased under low temperature stress does not necessary mean that exogenous H2S will increase endogenous H2S and cysteine content, nor protect the cell membrane or metabolic processes. To be clear, I am not claiming that it would not lead to those effects, particularly since the authors have the data that backs it up, but this sentence does not provide the information that would lead to that conclusion on its own. This should be rephrased to avoid logical mistakes.

-Response: Thank you very much for the reviewer's comments. According to the experimental results, we reorganized the language to express our views more accurately, and described L345-L354 in red font in the manuscript.

-7: L518. H2S does not induce “plant-pathogen interactions”, since there are no pathogens in this study. It does however induce the expression of genes related to plant-pathogen interactions. This should be rephrased.

-Response: Thanks very much for the reviewer's suggestion, we have adjusted the text in the manuscript and marked L408-L410 with red font.

-8: L521. I do not think that the growth of pepper plants was quantified in this study. If that the case, this should be reformulated ta make it clear that this was not an observation, but an educated guess.

-Response: Thank you very much for the reviewer's suggestions. We have revised the relevant descriptions in conclusion and abstract and marked them in red font in the manuscript.

-9: Range of

L125 Isn’t “relative conductivity” much more often referred to as “electrolytic leakage”?

Statistics should be explained in each figure description. What exactly is being compared to what.

-Response: Thanks very much for the reviewer's suggestion, we have modified L81 in the manuscript, and the relative conductivity has been changed to electrolytic leakage, statistical data of each graph has been added and marked with red font.

-10: In Figure 6E the ends of two of the columns occur in the middle of the “axis break”, and are therefore not visible.

-Response: Thanks very much to the reviewer's suggestion, we modified Figure 6 and reset the breakpoint.

We tried our best to improve the manuscript and made some changes in the manuscript. These changes will not influence the content and framework of the paper. And here we did not list the changes but marked in red in revised paper. We appreciate for Editors/Reviewers’ warm work earnestly, and hope that the correction will meet with approval.

Once again, thank you very much for your comments and suggestions.

Sincerely,

Song Xueping

College of Horticulture, Sichuan Agricultural University, Chengdu, Sichuan, China

Email address: [email protected]

Corresponding author: [email protected]

Reviewer 3 Report

Comments and Suggestions for Authors

The manuscript is prepared on a current and interesting topic, but some fundamental changes and additions must be made before publication: Introduction - on line 34 there is a reference (Pandel et al. 2023), which is supposed to document the importance of the study of resistance to cold. This is a "review" publication and I consider it appropriate to supplement this with other references that document specific studies in economically important plant species such as rice (Li et al. 2021 Czech Journal of Genetics and Plant Breeding /CJGPB/, 57 (4): 166-169; Zhang et al. 2022 CJGPB, 58(1):21-28, which prove the importance of studying at different stages of ontogenesis. Furthermore, in the Introduction section, on lines 52-66, the authors describe the importance of studying transcription factors and detection methods, it would be appropriate to add the importance of other transcription factors here, for example the reference Yu et al. 2022, CJGPB, 58(4):210-212, where it is documented including a functional analysis (GMO) and the same stress test used by the authors in this study was used, it is appropriate to include it in the Discussion section as well.

Material and Methods - here it is necessary to add subsections 2.3.1., 2.3.3. and 2.3.4. number of technical replicates! This is crucial information that is missing. The results - they are processed in text at a good level, but I see the processing of Figures as problematic, where a complete description (self-explanatory) is missing, i.e. explanation of all used abbreviations / e.g. Figure 8/, what do the individual lines of parameter variability mean, letters and their statistical value (probability), Figure 1A - scale is missing, etc. It is necessary to significantly work on this for all Figures! Similar explanatory legends, but the titles must be with the manuscript Supplements.

Based on the above facts and comments, I recommend the manuscript for publication after major revision and second review.

Author Response

November 11, 2023

Dear Editors and Reviewers:

Thank you for your letter and for the reviewers' comments concerning our manuscript entitled "Molecular regulatory mechanism of exogenous hydrogen sulfide in alleviating low-temperature stress in pepper seedlings "(ID: ijms-2692160). Those comments are all valuable and very helpful for revising and improving our paper, as well as the important guiding significance tour researches. We have studied comments carefully and have made correction which we hope meet with approval. Revised portion are marked in red in the paper. The main corrections in the paper and the responds to the reviewer's comments are as flowing:

Reviewer:

-1. on line 34 there is a reference (Pandel et al. 2023), which is supposed to document the importance of the study of resistance to cold. This is a "review" publication and I consider it appropriate to supplement this with other references that document specific studies in economically important plant species such as rice (Li et al. 2021 Czech Journal of Genetics and Plant Breeding /CJGPB/, 57 (4): 166-169; Zhang et al. 2022 CJGPB, 58(1):21-28, which prove the importance of studying at different stages of ontogenesis.

-Response: Thanks to the reviewers for their valuable comments, we redescribed this argument and selected a new reference [2]: Yanying Shi, Erjing Guo, Xue Cheng, Lizhi Wang, Xiaoguang Yang, Effects of chilling at different growth stages on rice pho-tosynthesis, plant growth, and yield. Environmental and Experimental Botany, 203(2022)105045. to describe the effects of low temperature stress on different stages of rice

-2. in the Introduction section, on lines 52-66, the authors describe the importance of studying transcription factors and detection methods, it would be appropriate to add the importance of other transcription factors here, for example the reference Yu et al. 2022, CJGPB, 58(4):210-212, where it is documented including a functional analysis (GMO) and the same stress test used by the authors in this study was used, it is appropriate to include it in the Discussion section as well.

-Response: I would like to express my heartfelt thanks to the reviewers for their valuable comments, we added a new reference [14]: Yu T, Zhou H, Liu Z, Zhai H, Liu Q. The sweet potato transcription factor IbbHLH33 enhances chilling tolerance in transgenic tobacco. Czech J. Genet. Plant Breed. 2022;58(4):210-222.

-3. Material and Methods - here it is necessary to add subsections 2.3.1., 2.3.3. and 2.3.4. number of technical replicates! This is crucial information that is missing.

-Response: I would like to express my heartfelt thanks to the reviewers for their valuable comments. In Materials and Methods, we have added subsections, and in the description of the last sentence of the experimental design, we state that our sample is all three biological replicates.

-4. The results - they are processed in text at a good level, but I see the processing of Figures as problematic, where a complete description (self-explanatory) is missing, ie. explanation of all used abbreviations / eg. Figure 8/, what do the individual lines of parameter variability mean, letters and their statistical value (probability), Figure 1A - scale is missing, etc. It is necessary to significantly work on this for all Figures! Similar explanatory legends, but the titles must be with the manuscript Supplements.

-Response: I would like to express my heartfelt thanks to the reviewers for their valuable comments, We have added an explanation of numeric and significant letter labels in red font in all image annotations. "Each data point represents the mean of the three repeated samples, and the result is the mean ± standard deviation. According to Duncan test, different letters indicated a statistically significant difference (p< 0.05)." The ruler for Figure 1A has also been added, already marked on the figure with white font and a horizontal line.

We tried our best to improve the manuscript and made some changes in the manuscript. These changes will not influence the content and framework of the paper. And here we did not list the changes but marked in red in revised paper. We appreciate for Editors/Reviewers’ warm work earnestly, and hope that the correction will meet with approval.

Once again, thank you very much for your comments and suggestions.

Sincerely,

Song Xueping

College of Horticulture, Sichuan Agricultural University, Chengdu, Sichuan, China

Email address: [email protected]

Corresponding author: [email protected]

Round 2

Reviewer 3 Report

Comments and Suggestions for Authors

The authors accepted all comments. Nevertheless, the manuscript contains minor formal errors that must be corrected before publication: line 42 - ... low temperature, right.... Low temperature; line 88 - Capsicum (italics); line 622 - not a complete reference for tracing; line 628 - not correct (and uniform) journal name; line 633 - authors are missing the correct "." and ",".

The manuscript can be accepted after minor revision.

Author Response

November 11, 2023

Dear Editors and Reviewers:

Thank you for your letter and for the reviewers' comments concerning our manuscript entitled "Molecular regulatory mechanism of exogenous hydrogen sulfide in alleviating low-temperature stress in pepper seedlings "(ID: ijms-2692160). Those comments are all valuable and very helpful for revising and improving our paper, as well as the important guiding significance tour researches. We have studied comments carefully and have made correction which we hope meet with approval. Revised portion are marked in red in the paper. The main corrections in the paper and the responds to the reviewer's comments are as flowing:

Reviewer:

-1. The authors accepted all comments. Nevertheless, the manuscript contains minor formal errors that must be corrected before publication: line 42 - ... low temperature, right.... Low temperature; line 88 - Capsicum (italics); line 622 - not a complete reference for tracing; line 628 - not correct (and uniform) journal name; line 633 - authors are missing the correct "." and ",".

-Response: Thanks very much for the reviewer's patient guidance and valuable comments. We carefully checked L42 and L88 in the manuscript, and adjusted the manuscript to the correct writing style. In L622, L628, and L633, the full reference format was added to the journal format, the issue name was modified, and the correct use of symbols between the authors' names was corrected.

We tried our best to improve the manuscript and made some changes in the manuscript. These changes will not influence the content and framework of the paper. And here we did not list the changes but marked in red in revised paper. We appreciate for Editors/Reviewers’ warm work earnestly, and hope that the correction will meet with approval.

Once again, thank you very much for your comments and suggestions.

Sincerely,

Song Xueping

College of Horticulture, Sichuan Agricultural University, Chengdu, Sichuan, China

Email address: [email protected]

Corresponding author: [email protected]
